

# Marine introgressions and Andean uplift have driven diversification in neotropical Monkey tree frogs (Anura, Phyllomedusinae)

Diego Almeida-Silva[1,2], Leonardo Matheus Servino[1,3], Matheus Pontes-Nogueira[1] and Ricardo J. Sawaya[1]

[1] Centro de Ciências Naturais e Humanas, Universidade Federal do ABC, São Bernardo do Campo, São Paulo, Brazil
[2] Miguel Lillo, Unidad Ejecutora Lillo, San Miguel de Tucumán, Tucumán, Argentina
[3] Instituto de Biociências, Universidade de São Paulo, São Paulo, São Paulo, Brazil

## ABSTRACT

The species richness in the Neotropics has been linked to environmental heterogeneity and a complex geological history. We evaluated which biogeographic processes were associated with the diversification of Monkey tree frogs, an endemic clade from the Neotropics. We tested two competing hypotheses: the diversification of Phyllomedusinae occurred either in a "south-north" or a "north-south" direction in the Neotropics. We also hypothesized that marine introgressions and Andean uplift had a crucial role in promoting their diversification. We used 13 molecular markers in a Bayesian analysis to infer phylogenetic relationships among 57 species of Phyllomedusinae and to estimate their divergence times. We estimated ancestral ranges based on 12 biogeographic units considering the landscape modifications of the Neotropical region. We found that the Phyllomedusinae hypothetical ancestor range was probably widespread throughout South America, from Western Amazon to Southern Atlantic Forest, at 29.5 Mya. The Phyllomedusines' ancestor must have initially diverged through vicariance, generally followed by jump-dispersals and sympatric speciation. Dispersal among areas occurred mostly from Western Amazonia towards Northern Andes and the South American diagonal of dry landscapes, a divergent pattern from both "south-north" and "north-south" diversification hypotheses. Our results revealed a complex diversification process of Monkey tree frogs, occurring simultaneously with the orogeny of Northern Andes and the South American marine introgressions in the last 30 million years.

# INTRODUCTION

Extending from the central portion of Mexico through the entire Central and South America (*Morrone, 2014*), the Neotropical region hosts the greatest biodiversity on Earth (*Myers et al., 2000*; *Antonelli & Sanmartín, 2011*). The environmental heterogeneity in the Neotropics associated with its complex geological history from the early Cenozoic has

Corresponding author
Leonardo Matheus Servino,
leonardo.servino@ib.usp.br

driven patterns of species diversification, contributing to high levels of species richness and endemism for several clades (*Antonelli, 2016*). Regarding the northern portion of South America, well-documented marine introgressions occurred from mid to the late Cenozoic (~25–5 million years ago–Mya), the so-called Pebas and Acre systems (*Hoorn et al., 2010*; *Salas-Gismondi et al., 2015*). Probably related to global sea-level fluctuations (*Hoorn, 1993*), both flooding processes turned the Western Amazonia into a lacustrine environment during the Miocene (23–7 Mya; *Hoorn et al., 2010*; *Salas-Gismondi et al., 2015*), affecting the Magdalena River delta, paleo-Orinoco, and proto-Amazonas River basins. Henceforth, Western Amazonia underwent drastic changes until the emergence of current fluvial systems, including flow changes of its main rivers (*Díaz de Gamero, 1996*; *Albert, Val & Hoorn, 2018*). Moreover, some orogenic processes also promoted important changes in the Neotropics. The accelerated uplift of the Eastern Cordillera of the Andes during the Miocene ~10–4 Mya; (*Hoorn, 1993*; *Gregory-Wodzicki, 2000*) led to changes in the climatic and sedimentary sources for Western Amazonia (*Insel, Poulsen & Ehlers, 2010*; *Poulsen, Ehlers & Insel, 2010*; *Hoorn et al., 2017*). The Andes uplift has also played a similar role in the southwestern part of the Neotropics (*Giambiagi, Alvarez & Spagnotto, 2016*; *Rodríguez Tribaldos et al., 2017*; *Sundell et al., 2019*).

During the Oligocene (~25 Mya) and late Neogene (~5–3 Mya), the Andean orogeny, in addition to climatic factors throughout the entire Neogene and Quaternary (*Garzione et al., 2006*; *Hoorn et al., 2020*), has been responsible for the rapid emergence of the South American "diagonal of open/dry landscapes" (DODL; *Zanella, 2011*; *Azevedo et al., 2020*), a dry corridor composed mostly by a savanna-like vegetation. As DODL expanded, a single large forest block has been separated into the Amazonian and Atlantic forests (*Costa, 2003*; *Sobral-Souza, Lima-Ribeiro & Solferini, 2015*; *Peres, Silva & Solferini, 2017*), the latter being southern confined by marine introgressions (*Hernández et al., 2005*; *Abello, Posadas & Ortiz Jaureguizar, 2010*). Consequently, ancestor lineages were confined to either the Amazon or Atlantic forests, resulting in several endemic clades to each region (*e.g.*, *Capurucho et al., 2018*; *Machado et al., 2018*; *de Sá et al., 2019*).

Most studies evaluated processes leading to biota diversification in the Neotropics in a local-scale approach, focusing on ecologically or geographically restricted groups (*e.g.*, *Smith et al., 2014*; *Werneck et al., 2015*; *Guarnizo et al., 2016*). Studies focused on widespread clades, on the other hand, could elucidate the role of multiple processes over space and deep time, contributing to a wider understanding of the macroevolutionary framework (*e.g.*, *Vicente et al., 2017*; *Hamdan et al., 2019*; *Prieto-Torres et al., 2019*; *Pontes-Nogueira et al., 2021*; *Serrano et al., 2024*), given the complex dynamics of biogeographic processes in the Neotropical region. Among anurans, this scenario fits well for Monkey tree frogs (Anura, Phyllomedusinae), a highly diverse subfamily that represents a clear case of autochthonous endemism, with its diversification occurring entirely within the Neotropics (*Faivovich et al., 2010*; *Brennan et al., 2023*). Comprising 67 species (*Frost, 2024*), Phyllomedusinae occurs from Argentina to Mexico (38° S to 27° N), encompassing various biomes such as tropical forests, grasslands, savannas, and deserts (*Duellman, Marion & Hedges, 2016*; *Frost, 2024*).

Systematics of the subfamily seems to be well defined regarding closely related frog groups. Except for phylogenetic analyses exclusively based on morphology (*Haas, 2003*; *Wiens et al., 2005*), phyllomedusines are consistently recovered as monophyletic and as a sister taxon to Pelodryadinae, a subfamily endemic from the Australo-Papuan region, both constituting subfamilies of Hylidae tree frogs (*Wiens et al., 2005*; *Frost et al., 2006*; *Faivovich et al., 2010*; *Pyron & Wiens, 2011*; *Duellman, Marion & Hedges, 2016*; *Jetz & Pyron, 2018*; *Dubois, Ohler & Pyron, 2021*). Phylogenetic relationships for some clades of Phyllomedusinae are also consistent in the most comprehensive phylogenetic approaches (*Faivovich et al., 2010*; *Pyron & Wiens, 2011*). Some discussion occurs regarding the early branching events in the group, since molecular approaches to phyllomedusine phylogeny show a low sampling for *Phrynomedusa* Miranda-Ribeiro, 1923, a rare genus known only from a few localities of the Serra do Mar and Serra da Mantiqueira ranges in the Atlantic Forest (*Baêta et al., 2016*).

Molecular estimates indicate that the split between phyllomedusines and pelodryadines occurred during the late Paleocene (~55 Mya), a time when both the Neotropics and Australo-Papuan regions were connected to Antarctica *via* a land bridge (*Duellman, Marion & Hedges, 2016*; *Van Den Ende, White & van Welzen, 2017*). Such estimates also suggest that the most recent common ancestor (MRCA) of Phyllomedusinae emerged during the late Eocene, following the appearance of the MRCA of Pelodryadinae. Diversification within Phyllomedusinae started in the Oligocene (*Duellman, Marion & Hedges, 2016*; *Jetz & Pyron, 2018*) as South America began to separate from Antarctica. However, the earlier diversification in phyllomedusines is still under debate. Some evidence suggests that the first lineage to diversify was *Phrynomedusa* Miranda-Ribeiro, 1923, a genus occurring throughout the southern Neotropics (Atlantic Forest domain; *Faivovich et al., 2010*; *Pyron & Wiens, 2011*). So, it is possible that the MRCA of all phyllomedusines would have diversified in a south-north direction. This pattern was already identified in some lineages that diversified in South America after their ancestral lineages arrived *via* Antarctica (*e.g.*, ungulate mammals, *Reguero et al., 2014*; orchids, *Givnish et al., 2016*; sea spiders, *Dietz et al., 2019*). Alternatively, some findings support *Cruziohyla* Faivovich et al., 2005, a Central American/Amazonian genus, as the first divergence in phyllomedusines (*Faivovich et al., 2010*; *Rivera-Correa et al., 2013*; *Portik et al., 2023*). Therefore, it is possible that the MRCA of all phyllomedusines was occupying northern regions of the Neotropics, diversifying in a north-south direction. Furthermore, certain speciation processes within phyllomedusines have tentatively been associated with the uplift of the Eastern Cordillera of the Andes (*Ron, Almendariz & Cannatella, 2013*; *Duellman, Marion & Hedges, 2016*). However, the historical biogeography of this clade has not yet been explored in a statistical framework.

Herein, we combined sequences of multiple molecular markers from 57 species of Phyllomedusinae to produce a time-calibrated phylogeny. We then reconstructed the subfamily's diversification throughout the Neotropical region. Firstly, we evaluated how the hypothetical ancestors of Phyllomedusinae have been distributed throughout the Neotropics. We tested two competing hypotheses: (1) the MRCA of all phyllomedusines occupied southern regions of the Neotropics, with extant lineages diversifying in a

south-north direction; and (2) the MRCA of all phyllomedusines occupied northern regions of the Neotropics, with extant lineages diversifying in a north-south direction. We evaluated which biogeographic processes must have driven the current subfamily distribution. We also tested the hypothesis that diversification in Phyllomedusinae was associated to marine introgressions and the Andean orogeny.

## MATERIALS AND METHODS

### Sequence data and phylogenetic analyses

We performed a phylogenetic inference using sequences from GenBank. We employed a single terminal for each species, aiming to minimize the utilization of more than one individual per species. Accordingly, we performed sequence selection based on identifying the voucher associated with the highest number of molecular markers for each species. Exceptions were made for species with limited molecular data, leading to the association of two or three vouchers with the same terminal. Within Phyllomedusinae, only 14 species were represented by two vouchers, while additional two required three vouchers (see Data S1 for details). Given the numerous taxonomic revisions and species descriptions within the group in recent years (*e.g.*, *Faivovich et al., 2010*; *Baêta et al., 2016*; *Castroviejo-Fisher et al., 2017*; *Pereira et al., 2018*; *Andrade et al., 2020*), we also prioritized sequences that included associated locality information (see Data S1). These localities were cross-referenced with available literature to ensure their congruence. Finally, sequences that do not align with our dataset in a given gene were checked using BLAST utility (*Zhang et al., 2000*), and we excluded the ones showing absence of query coverage.

Our analysis included 57 species of Phyllomedusinae in the ingroup, in addition to 20 Pelodryadinae and 18 Hylinae, both subfamilies comprising the outgroup. Also, our outgroup further included another 25 species from various frog families (*i.e.*, Bufonidae, Leptodactylidae, Odontophrynidae, Hemiphractidae, Ceratophryidae; Data S1), to consider recently recovered systematic relationships (*Feng et al., 2017*; *Jetz & Pyron, 2018*; *Hime et al., 2021*; *Portik et al., 2023*). Our molecular sampling covered 85% of all known species of Phyllomedusinae, including all genera of the subfamily (*Frost, 2024*). We comprised all the species of *Callimedusa Duellman, Marion & Hedges, 2016* (six spp.), *Cruziohyla Faivovich et al., 2005* (three spp.), *Hylomantis* (two spp.), and *Pithecopus* Cope, 1866 (12 spp.), as well as a representative selection of *Agalychnis* (nine species sampled from 14 spp. described), *Phasmahyla* Cruz, 1991 (seven species sampled from eight spp. described), *Phyllomedusa* Wagler, 1830 (15 species sampled from 16 spp. described), and *Phrynomedusa* (three species sampled from six spp. described). We searched for 13 molecular markers, both nuclear (CXCR4, POMC, RAG1, RHOD, SIAH, and Tyr) and mitochondrial genes (12S, tRNA-Val, 16S, tRNA-Leu, ND1, tRNA-Ile, and CytB) genes.

The amount of missing information ($\bar{x}$ = 33%, ranging from 6% to 92%; not accounting for gaps) should not seem alarming, considering that the two best-represented genes in our analyses (12S and 16S) provided a strong backbone for placing most species, as shown by *Pyron & Wiens (2011)*. In fact, 81% of the species had complete data for the 12S gene partition ($\bar{x}$ = 12% missing data), while the 16S partition was fully represented for 79% of the species ($\bar{x}$ = 9% missing data), and all species were represented in at least one of them.
Previous studies have supported this sample design for conducting model-based phylogenetic analyses, both theoretically and empirically (*e.g.*, *Wiens, 2003*; *Driskell et al., 2004*; *Thomson & Shaffer, 2010*; *Wiens & Morrill, 2011*), yielding taxonomically highly congruent and well-supported results (for a detailed discussion, see *Pyron & Wiens, 2011*). We used *MAFFT* by EMBL-EBI web toolkit (*Li et al., 2015*) for aligning our sequences. For coding markers, we employed the automatic strategy for alignment. For non-coding mitochondrial fragments, we used the Q-INS-i algorithm, to consider their secondary structure (*Katoh & Toh, 2008*).

Our complete dataset (8,660 bp, 120 terminals; Data S2) was divided into a set of 29 partitions. We set codon positions as separate partitions for the protein-coding genes (ND1, CytB, CXCR4, POMC, RAG1, RHOD, SIAH, Tyr), while 12S, 16S, and the transfer RNA molecular markers (tRNAVal, 16S, tRNALeu, and tRNAIle) were each set as a single partition. We performed model selection using bModelTest (transitionTransversionSplit model set; *Bouckaert & Drummond, 2017*), a Bayesian approach conducted concurrently with phylogenetic inference and node dating in the software *BEAST* v2.7.6 (*Bouckaert et al., 2019*). We conducted two independent Markov chain Monte Carlo (MCMC) simulations with a chain length of 150,000,000 generations and a pre-burn-in of 25% at the *CIPRES Science Gateway* (*Miller, Pfeiffer & Schwartz, 2010*). We linked the partitions into one phylogenetic species tree and kept the clock and site models unlinked.

To date our phylogeny using a fossil-calibrated phylogenomic tree, we followed *Hime et al. (2021)*. Specifically, we set a prior for the split between Phyllomedusinae and Pelodryadinae to the late Paleocene to early Eocene period (47.5 Mya; 95% CI: [42.0–53.4] Mya), a prior for the early diversification of Hylidae 58.9 Mya (95% CI: [54.4–64.2] Mya), and another two priors for both the early diversification of Hylinae (49.5 Mya; 95% CI: [44.9–54.4] Mya) and Hemiphractidae (43.9 Mya; 95% CI: [35.5–51.9] Mya) in the outgroup. We estimated the mean clock rates for all the other partitions under weak priors (1/x distribution). We inferred the species tree using a Yule model prior under a strict clock, while keeping all other priors at their default values. We assessed the convergence of the MCMC chains by examining the estimated sample size (ESS > 200) and checking for model parameter stationarity using *TRACER* 1.7 (*Rambaut et al., 2018*). We discarded the initial 25% of each chain as burn-in and summarized the output as a maximum clade credibility (MCC) tree (Data S3) using mean node heights in *TreeAnnotator* v.2.6.2 (*Bouckaert et al., 2014*). We pruned the MCC tree using the *ape* R package (*Paradis, Claude & Strimmer, 2004*), retaining only the species of Phyllomedusinae and Pelodryadinae for subsequent ancestral geographic range estimation.

## Geographic distribution data

Our geographic dataset consists of georeferenced points obtained from the Global Biodiversity Information Facility (*GBIF, 2024*; https://www.gbif.org/), which is the largest online source of distributional records (*Zizka et al., 2020*). We obtained the geographic points using the package *rgbif* (*Chamberlain & Boettiger, 2017*) in R software (*R Core Team, 2024*), resulting in 16,705 unique points for all species. We carefully reviewed geographic distributions in QGIS software (*QGIS.org, 2024*), comparing them with species

distribution presented in *Frost (2024)* and with specialized literature depicting geographic distributions (see Data S1 for details of all the literature used to filter our dataset and for our final geographic file).

## Study area and regionalization

Multiple regionalization schemes for the Neotropical region have been proposed in the literature (*e.g.*, *Olson et al., 2001*; *Morrone, 2006*, *2014*; *Dinerstein et al., 2017*; *Escalante, 2017*). Studies focusing on Neotropical species often involve a considerable number of biogeographic units due to its landscape heterogeneity (see *Carneiro et al., 2018*; *Réjaud et al., 2020*; *Pontes-Nogueira et al., 2021*). We defined 12 biogeographic units (Fig. 1A). We defined 11 units based on relevant landscape modifications that could have influenced Phyllomedusinae diversification (Fig. 1B), such as the uplift of mountain ranges (*e.g.*, Cordilleira of Andes), riverine barriers (*e.g.*, Amazonas and Madeira rivers), and phytophysiognomic differences (*e.g.*, DODL). All these landmarks follow previous regionalization schemes based on terrestrial ecoregions of the world (*Olson et al., 2001*; *Dinerstein et al., 2017*), so we decided that these 11 units were sufficient to capture the major landscape modifications that could have affected species diversification. Given that the Neotropics were once connected to Oceania through an Antarctic land bridge, we included the Australo-Papuan Pelodryadinae subfamily (the sister clade to the Phyllomedusinae subfamily) sampled in our phylogeny for estimating the ancestral geographic range. We also included Oceania in our regionalization scheme to refine the biogeographic results regarding the MRCA of Phyllomedusinae+Pelodryadinae, totaling 12 biogeographic units.

Therefore, our regionalization scheme encompasses the following regions (Fig. 1A): Central America (A), being southern limited by Chocó Department and the Pacific Coast of Colombia; its connectivity southwards has been enhanced over time due to the formation of the Isthmus of Panama, Northern Andes (B), encompassing Western, Central, and Eastern Cordilleras of the Northern Andes; it became a geographic barrier during the Miocene due to the acceleration on the uplift of the Eastern Cordillera, Western (C), Eastern (D), and Southern Amazonia (E), limited by the Amazonas and Madeira rivers; these three areas were differently affected by marine introgressions occurred during Miocene, Caatinga (F), reducing connectivity between forested areas as the DODL expanded, Cerrado (G), reducing connectivity between forested areas as the DODL expanded, Northern (H), Central (I), and Southern Atlantic Forest (J), divided by the Serra do Mar and Mantiqueira Mountain Ranges; these three areas were differently affected by the late uplift of both mountain ranges, Chaco/Pampas (K), encompassing Chaco, Pantanal, and the Uruguayan savanna, northern limited by Araucaria moist forests; reducing connectivity between forested areas as the DODL expanded, and Oceania (L), comprising the whole New Guinea island, the Wallacea region, and Australia; a continent with a complex history of connectivity with South America through Antarctica over the geological time.

**Figure 1** **A, Map of biogeographic units used in Neotropical region and Oceania, adapted from terrestrial ecoregions of the world (*Olson et al., 2001*; *Dinerstein et al., 2017*).** The areas are: Central America (A), Northern Andes (B), Western Amazonia (C), Eastern Amazonia (D), Southern Amazonia (E), Caatinga (F), Cerrado (G), Northern Atlantic Forest (H), Central Atlantic Forest (I), Southern Atlantic Forest (J), Chaco/Pampas (K), and Oceania (L); B, The summary of events acting as potential geographic barriers. We have used double arrows to specify instances where dispersal probabilities between the areas denoted were the only ones being affected; in contrast, we used commas to specify the cases where any dispersal through those areas had their probability decreased. Time stratification was applied to address landscape dynamics to our analysis, being indicated by the gray-white transition in geological time scale. In Neotropics, the complex Amazonas/Madeira was denoted as a geographic barrier to dispersal between Amazonian areas since the late-Miocene previously, Pebas and Acre systems were actuating on the same region along the entire Miocene. Moreover, Paranaense sea was another marine introgression occurring in Neotropics along the Miocene. The increase in connectivity between North and South America since the mid-Miocene, due to the formation of Panamá Isthmus bring another example of a geographic barrier "softened" through time. On the other hand, the uplift rates of Northern Andes and Serra do Mar and Mantiqueira Mountain Ranges had increased since the mid-Miocene, becoming a harsher barrier. This is also the case of DODL, that reduced connectivity between Amazonian and Atlantic forested areas by the expansion of aridity since the very late-Miocene. See Data S4 and Tables S4.1 and S4.2 for details.

## Ancestral geographic range estimation

Ancestral range estimation is performed based on the current distribution of sampled species and their phylogenetic relationships (*Sanmartín, 2016*). Several models for ancestral range estimation have been proposed in the literature, with the Dispersal-Vicariance Analysis (DIVA; *Ronquist, 1997*), the Dispersal-Extinction-Cladogenesis (DEC; *Ree et al., 2005*; *Ree & Smith, 2008*), and the BayArea model (*Landis et al., 2013*) being the most widely employed. These models have been implemented in *BioGeoBEARS* R package (*Matzke, 2013*, *2014*), which provides a unified Maximum Likelihood (ML) environment for biogeographic analyses. This allows for the use of parameters controlling biogeographic processes and model testing, eliminating the need for arbitrary model selection. We implemented 18 models in *BioGeoBEARS*, all of them being variations of DEC, DIVALIKE (the ML version of the original DIVA included in *BioGeoBEARS*), and BAYAREALIKE (the ML version of the original BayArea included in *BioGeoBEARS*; *Matzke, 2013*). Some models considered time-stratified dispersal matrices (TS), which are multipliers based on the landscape evolution of the study region (see Fig. 1B, and Supplementary Data S4 and Table S4.1 for details). The values in the TS matrices restrict the probabilities of dispersal between geographic units, ranging from 0 (when a geographic barrier completely prevents dispersal) to 1 (when there are no dispersal limitations between units). All TS models also included an areas-allowed matrix, informing which areas are allowed to be estimated according to their existence in certain timeframes. We suppressed the estimation of the Central American unit (A) prior to 23 mya because the Isthmus of Panama was not formed by this timeframe (see Discussion). To weigh the relative significance of the TS matrices, we also included the free parameter 'w' which acts as an exponent on the matrices (see *Dupin et al., 2017*). Additionally, to account for the colonization of novel biogeographic areas at the time of cladogenesis (*Matzke, 2014*, *2022*; *Klaus & Matzke, 2020*), we included the parameter 'j'. We set the maximum range size to five, which corresponds to the number of areas occupied by the most widespread species in our clade. We compared all the models using AIC (Akaike Information Criterion) and calculated Akaike weights (*Akaike, 1974*; *Burnham & Anderson, 2004*; *Wagenmakers & Farrell, 2004*).

## RESULTS

### Phylogeny and divergence time estimation

Bayesian inference recovered all the genera in our sample with high posterior probabilities (PP = 1.00 for all genera; Fig. 2; Data S5), both in Phyllomedusinae and Pelodryadinae. We recovered *Cruziohyla* as the sister clade to the other Phyllomedusinae genera (PP = 1.00), with the MRCA of the genus dating from 10.3 Mya (HPD 95%: 7.5–13.2 Mya). *Phrynomedusa*, the next diverging lineage (PP = 0.50), exhibited an MRCA that diversified from 13.7 Mya (HPD 95%: 10.8–16.5 Mya). We found *Agalychnis* Cope, 1864 as the sister to *Hylomantis* Peters, 1873 (PP = 0.99), a clade age estimated to be 21.7 Mya (HPD 95%: 20.1–23.6 Mya). Regarding the core of Phyllomedusinae (*i.e.*, MRCA of *Callimedusa*, *Phasmahyla*, *Phyllomedusa*, and *Pithecopus*; PP = 1.00), a clade mainly diversified in

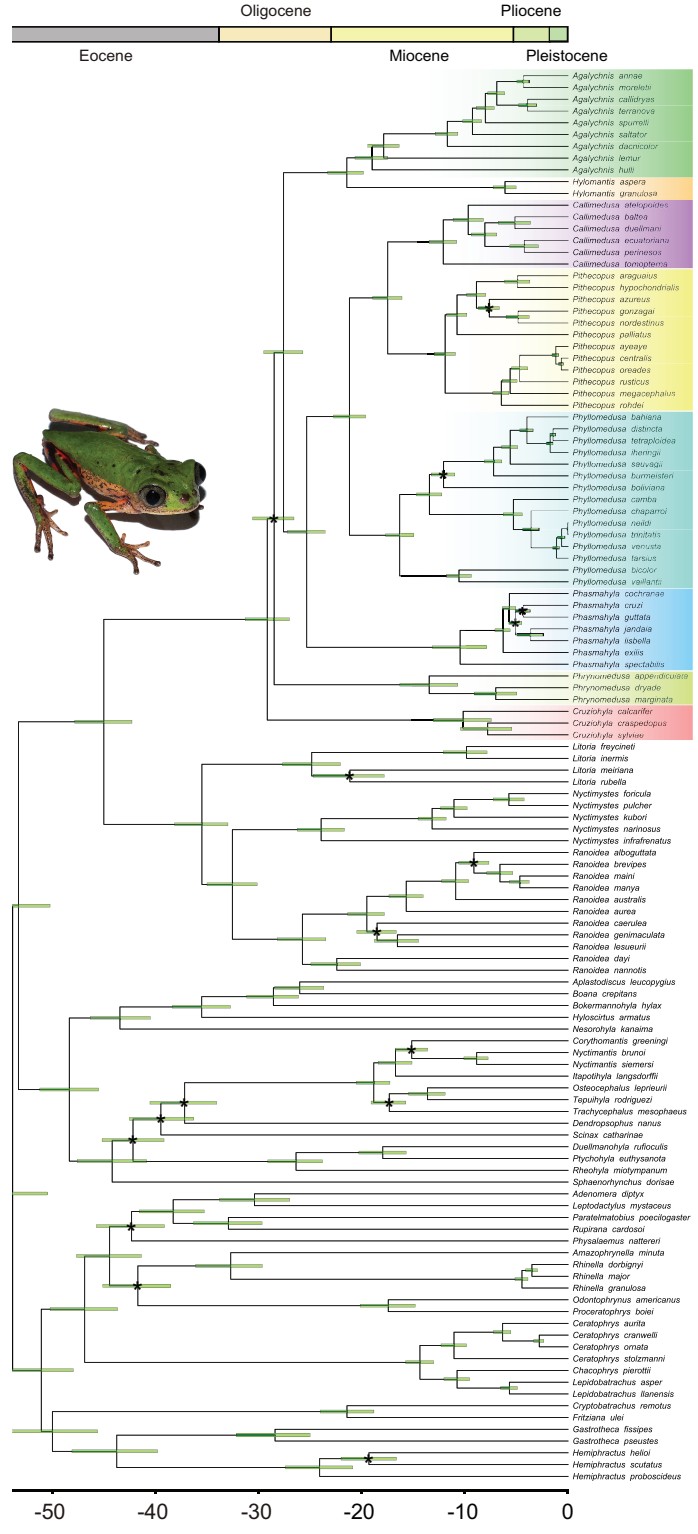

**Figure 2 Bayesian dated phylogenetic tree of Phyllomedusinae, based on 13 mitochondrial and nuclear concatenated loci (8,660 bp, 120 terminals).** Horizontal green bars represent the 95% HPD (height posterior density) intervals for the divergence date estimates. Black asterisks indicate clades where the value of posterior probability (pp) is lower than 0.9. See Data S1 for details about partitioning and the models of nucleotide substitution and Data S5 for detailed posterior probabilities. Photo credit (a specimen of *Pithecopus rohdei*): Henrique Silva Guedes Folly. 

**Table 1 AIC comparisons of the Ancestral range estimation models.**

| Models | LnL* | n* | d* | e* | j* | w* | AIC | AIC weights |
|---|---|---|---|---|---|---|---|---|
| **DECTS+j*** | **−206.7** | **3** | **0.01** | **<0.001** | **0.061** | **-** | **419.4** | **0.99** |
| BAYAREALIKETS+j | −211.7 | 3 | 0.0072 | <0.001 | 0.099 | - | 429.4 | 0.0065 |
| DIVALIKETS+j | −211.8 | 3 | 0.012 | <0.001 | 0.047 | - | 429.5 | 0.0063 |
| DECTS | −217.9 | 2 | 0.013 | <0.001 | - | - | 439.8 | <0.001 |
| DECTS+j+w | −220.9 | 4 | 0.0073 | <0.001 | 0.027 | 0.094 | 449.9 | <0.001 |
| BAYAREALIKETS+j+w | −223.3 | 4 | 0.0044 | <0.001 | 0.052 | 0.079 | 454.6 | <0.001 |
| DIVALIKETS+j+w | −226.7 | 4 | 0.0068 | <0.001 | 0.022 | 0.047 | 461.4 | <0.001 |
| DIVALIKETS | −230.3 | 2 | 0.01 | 0.01 | - | - | 464.5 | <0.001 |
| DECTS+w | −232.6 | 3 | 0.0073 | <0.001 | - | 0.0031 | 471.3 | <0.001 |
| DIVALIKETS+w | −237 | 3 | 0.0089 | <0.001 | - | 0.0039 | 480 | <0.001 |
| BAYAREALIKE+j | −249.5 | 3 | 0.0023 | <0.001 | 0.025 | - | 504.9 | <0.001 |
| DEC+j | −249.7 | 3 | 0.0035 | <0.001 | 0.012 | - | 505.4 | <0.001 |
| DEC | −256.3 | 2 | 0.0041 | <0.001 | - | - | 516.6 | <0.001 |
| DIVALIKE+j | −255.6 | 3 | 0.0038 | <0.001 | 0.012 | - | 517.2 | <0.001 |
| BAYAREALIKETS | −257.2 | 2 | 0.013 | 0.053 | - | - | 518.5 | <0.001 |
| DIVALIKE | −264.3 | 2 | 0.005 | <0.001 | - | - | 532.5 | <0.001 |
| BAYAREALIKETS+w | −267.7 | 3 | 0.0069 | 0.053 | - | 0.0026 | 541.3 | <0.001 |
| BAYAREALIKE | −283.6 | 2 | 0.0038 | 0.055 | - | - | 571.2 | <0.001 |

Note:
*LnL = log-likelihood of the model. n = number of free parameters in the model (that being d, e, j and w); d = rate of range expansion (*i.e.*, anagenetic dispersal); e = rate of range contraction (*i.e.*, extinction); j = jump dispersal process; and w = dispersal multiplier parameter (for TS models). **DECTS+j is shown in bold and represents the best model under AIC and AIC weights.

forested areas, our estimation suggests an age of 25.7 Mya (HPD 95%: 23.8–27.5 Mya). We recovered *Phasmahyla* as the sister group to the other three genera, with the MRCA of the clade *Phyllomedusa* (*Callimedusa* + *Pithecopus*) (PP = 1.00) estimated to be 21.5 Mya (HPD 95%: 19.9–23.0 Mya).

### Ancestral geographic range estimation

The best-fitted model in our analysis was DECTS+j (Table 1; AIC = 419.4; AICw ~0.99), incorporating landscape evolution in the Neotropics and jump-dispersal processes (Fig. 3; see Data S6 for details). We found the divergence of Phyllomedusinae + Pelodryadinae (45.5 Mya; HPD 95%: 42.7–48.4 Mya) occurring through vicariance, with the MRCA of Pelodryadinae subsequently dispersing throughout Oceania (unit L; Figs. 3 and 4A; Data S5). Meanwhile, the MRCA of Phyllomedusinae was initially occupying Western Amazonia (unit C), expanding its range to Northern Atlantic Forest (units CH; Figs. 3, 4A).

The earliest diversification event within Phyllomedusinae occurred when a vicariant process occurred at its MRCA, isolating the *Cruziohyla* ancestor in Western Amazon (unit C, 29.5 Mya; HPD 95%: 27.3–31.6 Mya; Fig. 4B). Subsequently, a jump-dispersal process would be responsible for the populations in the Central Atlantic Forest at 28.8 Mya (HPD 95%: 26.9–30.9 Mya), leading to the origin of the *Phrynomedusa* genus (unit I; Figs. 3;

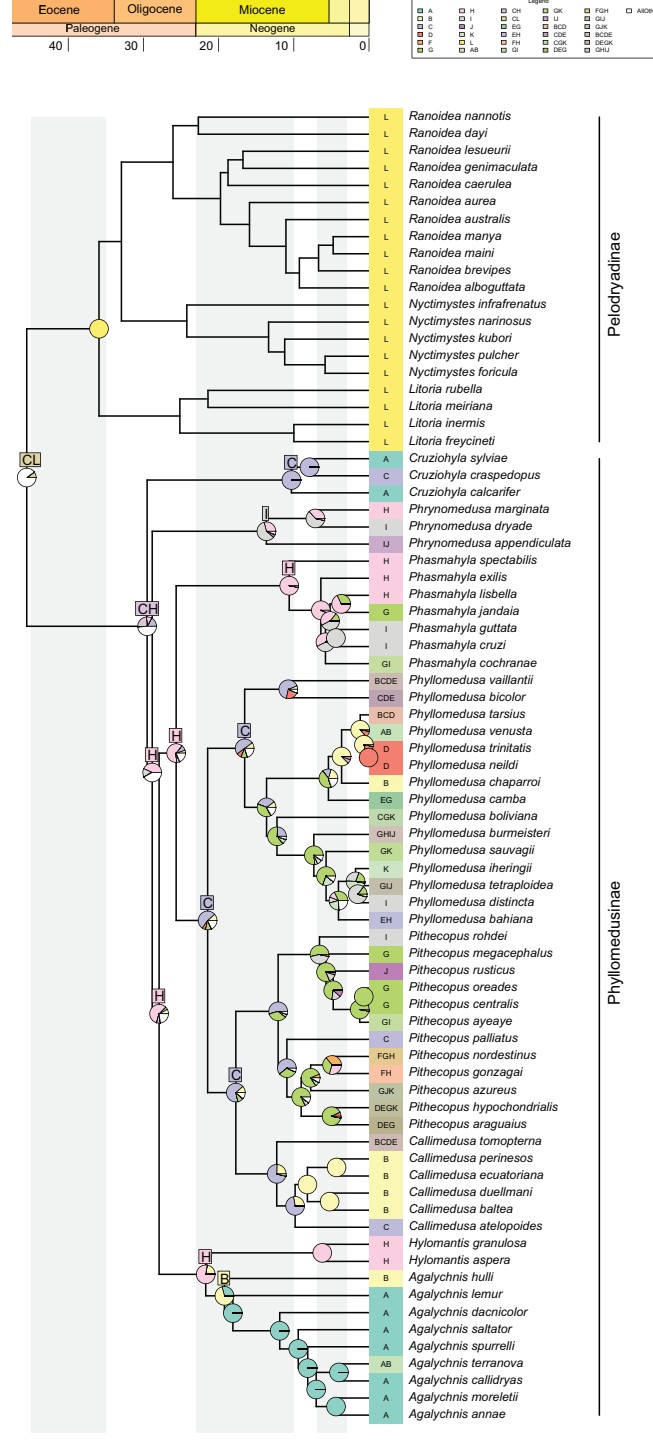

**Figure 3  Ancestral range estimations from DECTS+j model implemented in BioGeoBEARS.** Colored boxes with letters represent the most probable range estimated by the model (boxes for nodes in the last 20 million years are not shown for better visualization of these nodes). Ancestral area estimations at nodes represent areas before an inferred instantaneous speciation event; colored rectangles at the tips with letters indicate the current distribution of extant species. Pie charts on nodes represent probabilities of ranges. Only the four most probable states at each node are shown for better visualization, and blank spaces represents all other probabilities (see Data S5 for more information).

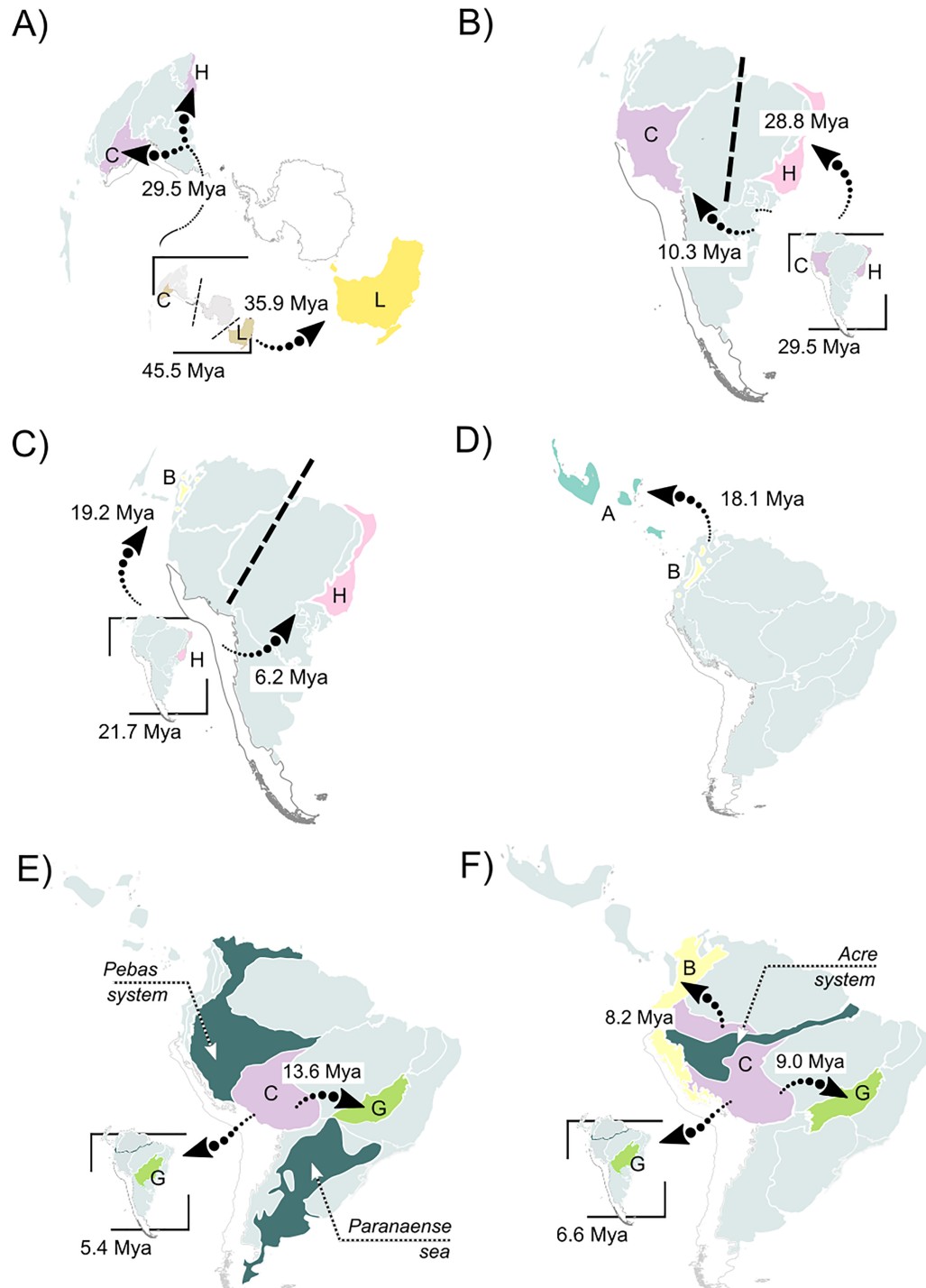

**Figure 4 Summary of the recovered biogeographic processes using arrows, following our time stratification (from the oldest to the newest).** (A) A vicariant event at the split between Pelo-dryadinae + Phyllomedusinae, isolating both subfamilies from a wide-distributed MRCA; (B) a vicariant event originating the *Phrynomedusa* genus by the isolation of populations in the Central Atlantic Forest, the same biogeographic unit from where jump-dispersed the populations who expanded into the Western Amazonia and lead to the MRCA of (*Phyllomedusa* (*Callimedusa* + *Pithecopus*)) at 32.4 Mya; (C) a vicariant event, resulting in the emergence of *Hylomantis* and *Agalychnis* genera from a wide-distributed MRCA; (D) another vicariant event at the early diversification of *Agalychnis* genus, at the divergence of

**Figure 4** (continued)
the Andean species *A. hulli*; (E) north-south pattern of diversification of *P. burmeisteri* group, whose MRCA jump dispersed from Western Amazonia giving rise to a diversification along Chaco and, subsequently, Atlantic Forest' units during the early opening of DODL; (F) the divergent patterns of diversification in *Pithecopus* (north-south, colonizing the Cerrado by jump-dispersal in two waves) and *Callimedusa* (south-north, colonizing the Northern Andes by range expansion) genera, both taking place in the Miocene concurrently to the Pebas system. See Fig. 3 for info about the units' colors and letters. Maps adapted from terrestrial ecoregions of the world (*Olson et al., 2001*; *Dinerstein et al., 2017*) to highlight geological events.                                                 

Data S6). Ancestral populations underwent speciation processes in sympatry at 21.7 Mya (HPD 95%: 20.1–23.6 Mya; Fig. 4C), resulting in the emergence of *Hylomantis* in the Northern Atlantic Forest (unit H). By the same time, the MRCA of *Agalychnis* reached Northern Andes (unit B; Fig. 4C) and, later, Central America (A), through jump-dispersal processes (Fig. 4D).

The MRCA of (*Phasmahyla*, (*Phyllomedusa*, (*Callimedusa*, *Pithecopus*))) remained in Northen Atlantic Forest (unit H; 25.7 Mya; HPD 95%: 23.8–27.5 Mya; Fig. 4B). From this ancestor, a process in sympatry was responsible for the emergence of the *Phasmahyla* genus, while a jump-dispersal to Western Amazonia (unit C; Fig. 4B) led to the MRCA of (*Phyllomedusa*, (*Callimedusa*, *Pithecopus*)). We identified at least three major diversification patterns within the *Phyllomedusa* genus (Fig. 3). Firstly, the clade (*Ph. vaillanti*, *Ph. bicolor*) originated at 10.7 Mya (HPD 95%: 9.5–11.9 Mya) through sympatric speciation within Western Amazonia, followed by subsequent dispersal throughout Amazonia and the Northern Andes. Jump-dispersal events are predominant in the diversification of the other species of the genus, sharing an ancestor that initiated the colonization of the Cerrado at 13.6 Ma (HPD 95%: 12.4–14.9 Ma). From this ancestor, the second diversification pattern occurred in the *Ph. burmeisteri* Boulenger, 1882 group underwent jump-dispersal events, colonizing Cerrado (unit G; Fig. 4E), followed by subsequent dispersal to the Atlantic Forest units (H, I, and J units), Chaco (unit K), and Southern Amazon Forest (unit E) from 13.6 Mya (HPD 95%: 12.4–14.9 Mya) to the present. More recently, the *Ph. tarsius* Cope, 1868 group underwent a third diversification wave of *Phyllomedusa*, where jump-dispersal events later moved north to Amazonia, Northern Andes, and Central America from 5.4 Ma (HPD 95%: 4.5–6.4 Ma) onwards.

The MRCA of (*Pithecopus*, *Callimedusa*) remained in Western Amazonia (unit C) at 17.7 Mya (HPD 95%: 16.3–19.2 Mya), as well as the ancestors for both genera. *Pithecopus* displayed a colonization pattern in the Cerrado region (unit G) through two separate jump-dispersal events. The first jump-dispersal to the Cerrado occurred at 12.1 Mya (HPD 95%: 11.1–13.1 Mya; Fig. 4F), leading to a clade that diversified through sympatric speciation in Cerrado and additional jump-dispersals to the Atlantic Forest (units I and J). The second jump-dispersal to the Cerrado took place around 9.0 Mya (HPD 95%: 8.1–10.0 Mya; Fig. 4F), resulting in a clade primarily diversifying through sympatric speciation in Cerrado and range expansions (*i.e.*, anagenetic dispersals) to the Atlantic Forest (units H and J) and Chaco (unit K). Additionally, the clade (*Pi. hypochondrialis*, *Pi. araguaius*)

reached the Northern/Eastern Amazonia. The MRCA of (*Pi. gonzagai, Pi. nordestinus*) originated from a jump-dispersal event from the Cerrado to Caatinga at 4.9 Mya (HPD 95%: 3.8–6.1 Mya). On the other hand, the *Callimedusa* genus had an early sympatric speciation that gave rise to *C. tomopterna* Cope, 1868 at 12.3 Mya (HPD 95%: 10.9–13.6 Mya). This species expanded its range throughout the entire Amazonian region (units C, D, and E) and Northern Andes (unit B). Subsequently, another sympatric speciation occurred within Western Amazonia around 9.8 Mya (HPD 95%: 8.3–11.3 Mya; Fig. 4F). This cladogenesis resulted in the origin of *C. atelopoides* Duellman, Cadle, and Cannatella, 1988 and the MRCA of the other *Callimedusa* species, which reached the Northern Andes by jump-dispersal (Fig. 4F), afterward diversifying in sympatry since then.

## DISCUSSION

In the present study, we provided a detailed analysis of the diversification and colonization history of Monkey tree frogs across the Neotropics. By sampling 85% of the formally described phyllomedusine species, using mitochondrial and nuclear markers, our results represent a robust framework to discuss the processes concerning the biogeographic history of the group. Regarding the phylogenetic relationships among phyllomedusine genera, our topology is mostly congruent with previous studies in the literature based on molecular data (*Faivovich et al., 2005*, *2010*; *Pyron & Wiens, 2011*; *Rivera-Correa et al., 2013*; *Duellman, Marion & Hedges, 2016*; *Jetz & Pyron, 2018*). Also, our results on node dating are very similar to those found by previous studies (*Feng et al., 2017*; *Hime et al., 2021*; *Portik et al., 2023*). Combined with our results on ancestral geographic range estimation, node dating indicates that the diversification of phyllomedusines was markedly influenced by environmental changes resulting from the Miocene marine introgressions and Andean orogeny. Specifically, most lineages in Phyllomedusinae diversified in a dynamic scenario in Western Amazonia, which was limited by the Pebas System and the Paranaense sea (north/westward and southward, respectively) during the Miocene.

The most evident divergence recovered in phylogenetic inference was the relative position of the *Cruziohyla* and *Phrynomedusa* genera. Previous studies usually place *Phrynomedusa* as the first branching lineage in Phyllomedusinae (*Faivovich et al., 2005*; *Pyron & Wiens, 2011*) or composing a clade (*Cruziohyla*, *Phrynomedusa*) (*Duellman, Marion & Hedges, 2016*; *Jetz & Pyron, 2018*). Since the split between the lineage given rise to *Cruziohyla* and the other phylomedusines occurred at 29.5 Mya (HPD 95%: [27.3–31.7] Mya), our results imply a first colonization of the northern Neotropics during the Oligocene.

At the genus level, the topology we found for *Phasmahyla* differs from previous studies, but they also mostly disagree (*Faivovich et al., 2010*; *Pyron & Wiens, 2011*; *Duellman, Marion & Hedges, 2016*; *Jetz & Pyron, 2018*; *Pereira et al., 2018*; *Portik et al., 2023*). *Phasmahyla jandaia* Bokermann & Sazima, 1978 has frequently been identified as closely related to other species within the genus (*Faivovich et al., 2010*; *Pyron & Wiens, 2011*; *Rivera-Correa et al., 2013*; *Duellman, Marion & Hedges, 2016*; *Portik et al., 2023*), potentially composing a clade with *P. lisbella* Pereira et al., 2018 (*Portik et al., 2023*). We recovered the clade (*P. jandaia*, *P. lisbella*), but placed it within the *Phasmahyla* core.

Similarly, the identified clade (*P. guttata*, *P. cruzi*) is widely documented in the literature. However, even these relationships are not in line with the findings of the study boasting the highest representativeness in terms of the number of specimens and species for *Phasmahyla* (*Pereira et al., 2018*). It is important to note, however, that this paper used only one molecular marker (16s; see *Pereira et al., 2018* for further information). Also, some inconsistencies regarding *Phasmahyla* argue for a careful taxonomic review. *Phasmahyla cruzi* Carvalho-e-Silva, Silva & Carvalho-e-Silva, 2009, for instance, is assumed to be known only from its type locality (Rio das Pedras Reserve, Municipality of Mangaritiba, state of Rio de Janeiro, Brazil; *Frost, 2024*). However, there are no molecular data on GenBank from this population, and the species was represented by a specimen assigned to another locality (Picinguaba, municipality of Ubatuba, state of São Paulo, Brazil; *Faivovich et al., 2005*, *2010*; *Pyron & Wiens, 2011*; *Duellman, Marion & Hedges, 2016*; *Pereira et al., 2018*; present study, Data S1). Hence, we contend that the phylogenetic relationships within *Phasmahyla* remains subject to debate. We found other occasional divergences regarding the relative position of some species in *Pithecopus* and *Phyllomedusa*, compared to other studies (*Faivovich et al., 2010*; *Pyron & Wiens, 2011*; *Duellman, Marion & Hedges, 2016*; *Jetz & Pyron, 2018*; *Pereira et al., 2018*).

The split we found for the MRCA of Phyllomedusinae + Pelodryadinae occurred by vicariance (Figs. 3 and 4A; Data S6) in the early Eocene. Our results suggest that the MRCA of Phyllomedusinae + Pelodryadinae was widely distributed throughout South America, Oceania, and supposedly, Antarctica (Fig. 4A). The diversification between the two subfamilies during the passage from the late Eocene to the early Oligocene occurred concurrently with the convoluted process of separation between the three continents, as proposed by *Duellman, Marion & Hedges (2016)*. Hence, the vicariance associated was probably caused by landmass movements, promoting the initial divergence of pelodryadines and phyllomedusines. Moreover, our results emphasize that much of the early diversification of the Phyllomedusinae was influenced by this widespread South American ancestor, especially for the *Cruziohyla* and *Agalychnis* genera. Hence, although some phyllomedusine lineages had a "north-southern" diversification trend (see below), our results reject both the "south-north" and "north-south" diversification hypotheses for the Phyllomedusinae subfamily in general, as the diversification within the group occurred from Western Amazonia and Northern Atlantic Forest towards Northern Andes and the diagonal of dry landscapes. Our findings show a different scenario from *Duellman, Marion & Hedges (2016)*, where the authors argued that the split between the two subfamilies took place within the Neotropics, with "protopelodryadines" dispersing to Australia afterward. Since our sampling focused on Phyllomedusinae, it is difficult to extend the discussion to the context of the whole Hylidae family. We encourage future biogeographic studies to examine the question in more detail.

Our results from the DECTS+j model suggest that Western Amazonia (unit C) acted as a species pump (see *Rangel et al., 2018*) for the *Phyllomedusa*, *Pithecopus* and *Callimedusa* genera. This area is frequently identified as one of the ancestral ranges in several animal groups (*e.g.*, lizards, *Prates et al., 2017*; snakes, *Dal Vechio et al., 2018*; *Pontes-Nogueira et al., 2021*). Western Amazonia was recovered as the ancestral area for the clade

(*Phyllomedusa*, (*Callimedusa*, *Pithecopus*)) after a jump-dispersal in the split between *Phasmahyla* and that clade. This scenario took place from the late Oligocene to the middle Miocene, concomitantly with the occurrence of lacustrine conditions due to the Pebas system (*Hoorn et al., 2010*; *Jaramillo et al., 2010*; Figs. 4E and 4F), which profoundly affected the entire Western Amazonia. Previous studies also emphasize the influence of the Pebas system over the biogeography of neotropical anuran fauna, shaping their evolutionary history in different ways (*Carvalho, 1954*; *Zimmermann & Zimmermann, 1988*; *Fouquet et al., 2012*, *2021a*, *2022*; *Réjaud et al., 2020*). The absence of an overall pattern in Anura seems to be related to the diversity of natural history traits. Hence, the Pebas system turned Western Amazonia an unsuitable environment for ground-dwelling frogs, negatively affecting the diversification of terrestrial (Phyzelaphryninae and *Allobates*; *Zimmermann & Zimmermann, 1988*; *Fouquet et al., 2012*; *Réjaud et al., 2020*) and burrowing (*Synapturanus*; *Carvalho, 1954*; *Fouquet et al., 2021a*) clades. On the other hand, the marine incursion was crucial for the origin and diversification of aquatic clades (*Pipa*; *Fouquet et al., 2022*). Here, we found evidence of Pebas system promoting the diversification of arboreal anurans, given that the most speciose clade in Phyllomedusinae has originated and started to diversify in this area. This is an important finding, since other studies regarding arboreal frogs have shown diversification occurring after the regression of the Pebas system (*e.g.*, *Boana albopunctata* Spix, 1824 species group; *Fouquet et al., 2021b*). As far as we know, our present study provides the most comprehensive framework on this issue.

From Western Amazonia, we found that phyllomedusines expanded into the central and southern regions of the Neotropics in the last 16 million years (Figs. 3, 4E and 4F), encompassing the genera *Pithecopus* and *Phyllomedusa*. *Pithecopus* lineages jump-dispersed from Western Amazonia twice during the middle Miocene, occupying areas that would later become the "diagonal of open/dry landscapes" (DODL). These results are concomitant with previous biogeographic studies involving this genus (*Magalhães et al., 2024*). We observed a similar pattern in the *Ph. burmeisteri* group (*sensu Faivovich et al., 2010*), the most speciose clade in *Phyllomedusa*. Both cases exemplify a "north-southern" biogeographic colonization of the Neotropics, mentioned in the literature for several groups (snakes, *Wüster et al., 2002*; *Hamdan et al., 2019*; termites, *Carrijo et al., 2020*), including frogs (*e.g.*, *Fouquet et al., 2012*, *2014*). However, we could not extend this geographically oriented pattern of diversification to all phyllomedusines, as it seemed to occur only at lower taxonomic levels in Phyllomedusinae.

By occurring from the middle to late Miocene, the biogeographic patterns we found are concurrent with the transition from the Pebas to the Acre system, taking place in an Amazonian wetland. Firstly, all these lineages left Western Amazonia coincidentally with a period of recurring marine incursions of the Paranaense Sea (*Hernández et al., 2005*), which may have limited the Amazonia region southward (Figs. 4E and 4F). These findings suggest a progressive isolation of the core Phyllomedusinae in northern South America, agreeing with previous studies regarding the influence of the Paranaense Sea on the biogeographic history of South American herpetofauna (*e.g.*, *Seger et al., 2021*; *Abreu-Jardim et al., 2023*). The subsequent regression of the Paranaense Sea could have facilitated

the colonization of southern areas, helping to explain the diversification pattern we found. Secondly, the subsequent colonization of South American Atlantic Forest by some of these Miocene lineages of Phyllomedusinae is congruent with the very early opening of the DODL, and the late orogeny of the Serra do Mar Mountain Range. This pattern represents a jump-dispersal response to the retraction of forested areas in due to the environmental changes driven by the opening of DODL, commonly found in other anuran groups (*Pirani et al., 2020*; *Carvalho et al., 2021*). The whole biogeographic process also sheds light on our findings about the diversification of the *Phasmahyla* genus, which remained isolated in the South American Atlantic Forest in the last 10 million years.

Also, we found that several phyllomedusine lineages diversified along the Miocene from Western Amazonia northward (Figs. 3 and 4F), through the Andes and Central America. The biogeographic history of these species suggests a colonization of Northern Andes, occupying the forested lowlands from the Miocene to the Pleistocene. During a period of drastic changes in local drainage patterns due to the transition from the Pebas to the Acre system (*Hoorn et al., 2010*), we found that the *Phyllomedusa* genus has been particularly successful in colonizing the entire Amazonian region. Together, these results seem to reinforce the idea that phyllomedusine frogs were able to survive in the lacustrine environment resulting from changes in the Amazonian drainage pattern compared to other frog groups. Previous studies have already suggested that the isolation and geographic expansion in other arboreal frogs may have been affected by the Miocene marine introgressions, depending on their capacity to exploit wetland environments for reproduction (*e.g.*, *Ortiz et al., 2023*).

Following this northward diversification (Fig. 4F), we found some aspects of the historical biogeography of phyllomedusines closely related to the uplift of the Northern Andes. Populations of the MRCA of *Callimedusa* in Western Amazonia experienced an early divergence within this region, giving rise to *Callimedusa tomopterna* Cope, 1868 (Fig. 3). Subsequently, populations from the MRCA of other *Callimedusa* species jump-dispersed to the Northern Andes, where mountain uplifts potentially facilitated sympatric speciation during the mid-Miocene (Fig. 4F). The diversification of *Pithecopus* in the Cerrado can also be directly attributed to the Andes uplift. Although the MRCA of *Pithecopus* primarily was in Western Amazonia, ancestral populations reached the Cerrado by two separate jump-dispersal events (Fig. 3). Divergences coincided with the final opening of the DODL, where the late Andean uplift contributed to the uplift and dryness of the Brazilian Plateau and the subsidence of the Chaco region (*Zanella, 2011*; see also *Silva, 1995*; *Pontes-Nogueira et al., 2021*). Once ancestral lineages of *Pithecopus* colonized the Cerrado, they diversified in sympatry alongside the changing landscape of this region (Fig. 3). Thus, the uplift of the Andes probably played a main role in cladogenetic processes in phyllomedusine lineages during mountain uplift events, as well as in areas that underwent landscape changes influenced by the elevation of the mountain range.

The diversification of the *Agalychnis* (Figs. 3 and 4D) and *Cruziohyla* (Fig. 3) genera during their expansion into Central America is intriguing. In *Agalychnis*, the colonization of Central America by the ancestors of the Phyllomedusinae frogs during the Miocene

precedes the formation of the Isthmus of Panama, proposed to have occurred in the Plio-Pleistocene (~3 Mya; Fig. 3; *Haug & Tiedemann, 1998*; *O'Dea et al., 2016*). *Bacon et al. (2015)* demonstrated two significant waves of dispersal between South and North America at around 20 and 5 Mya, also preceding the recent formation hypothesis. At first glance, our results are in accordance with the first wave of dispersal, as the MRCA of *Cruziohyla* is synchronous with the very early formation of the Isthmus of Panama and early uplift of the Eastern Cordillera of Northern Andes (*Gregory-Wodzicki, 2000*), and the jump-dispersal present in the MRCA of *Agalychnis* is suggested to have occurred at the same time as this wave of dispersal according to our results. Recent biological (*Bacon et al., 2013*, *2015*; *Bloch et al., 2016*) and geological (*Farris et al., 2011*; *Montes et al., 2012*, *2015*; *Jaramillo et al., 2017*) findings suggest an older formation for the Isthmus of Panama (early to middle Miocene), despite divergent findings (*e.g.*, *O'Dea et al., 2016*). However, it is thought that range expansion (*i.e.*, anagenetic dispersal) is more plausible to occur by land (for land animals) and that jump-dispersals are predominantly associated with geographic barriers (see *Matzke, 2014* for more information), and the presence of these jump-dispersals could support the idea of a later formation of the Isthmus rather than with the earlier emergence hypothesis. Therefore, further studies considering the whole Hylidae family may address this issue more properly. The biogeography of the Neotropics is surely intriguing and intricate, and the history of the Monkey tree frogs described here adds another level of certainty to this statement.

## CONCLUSIONS

We found that the biogeographic history of Phyllomedusinae started with a vicariance splitting the Neotropical region, Oceania, and Antarctica. Indeed, vicariance was a common biogeographic process during the early diversification of phyllomedusines, while jump-dispersals are likely to have been responsible for the majority of colonizations in the group since the Miocene. Western Amazonia may have served as a species pump for most of the Monkey tree frogs, with species colonizing the area and diversifying sympatrically even during the highly unstable environment of the Miocene. Also, the orogeny of the Northern Andes should have played an important role in species diversification, promoting sympatric speciation both through the uplift of mountains and in areas with drastic landscape changes provoked by the elevation of the Andean Mountain range. Our results also reject both the "south-north" and "north-south" diversification hypotheses for Phyllomedusinae, although we observed some geographically oriented diversification at lower taxonomic levels. In brief, we have provided a comprehensive overview of the historical biogeography of this speciose group, enabling a highly detailed description of the diversification of this charismatic frog subfamily.

## ACKNOWLEDGEMENTS

We are grateful to Pedro Paulo Goulart Taucce, Tiago Fernandes Carrijo, and Marcelo José Sturaro for their outstanding contributions on the systematics and node dating issues. MPN also thanks Felipe Grazziotin for his help with the R software. We thank the three anonymous reviewers for their comments and suggestions that helped to improve the

manuscript. We also thank Henrique Folly for providing the photo presented on Fig. 2. LMS dedicates this study in honor of Marco Antonio Servino.

### Funding

This study was supported by Fundação Coordenação de Aperfeiçoamento de Pessoal de Nível Superior (Finance Code 001), Fundação de Amparo à Pesquisa do Estado de São Paulo (FAPESP; grants 20/12658-4, 21/10039-8, 22/05543-1 and 23/01785-3), Conselho Nacional de Desenvolvimento Científico e Tecnológico (CNPq; grant 307956/2022-9) and Consejo Nacional de Investigaciones Científicas y Técnicas (CONICET). The funders had no role in study design, data collection and analysis, decision to publish, or preparation of the manuscript.

### Grant Disclosures

The following grant information was disclosed by the authors:
Fundação Coordenação de Aperfeiçoamento de Pessoal de Nível Superior: Finance Code 001.
Fundação de Amparo à Pesquisa do Estado de São Paulo: FAPESP; grants 20/12658-4, 21/10039-8, 22/05543-1 and 23/01785-3.
Conselho Nacional de Desenvolvimento Científico e Tecnológico: CNPq; grant 307956/2022-9.
Consejo Nacional de Investigaciones Científicas y Técnicas: CONICET.

### Competing Interests

The authors declare that they have no competing interests.

### Author Contributions

- Diego Almeida-Silva conceived and designed the experiments, performed the experiments, analyzed the data, prepared figures and/or tables, authored or reviewed drafts of the article, and approved the final draft.
- Leonardo Matheus Servino conceived and designed the experiments, performed the experiments, analyzed the data, authored or reviewed drafts of the article, and approved the final draft.
- Matheus Pontes-Nogueira conceived and designed the experiments, performed the experiments, analyzed the data, prepared figures and/or tables, authored or reviewed drafts of the article, and approved the final draft.
- Ricardo J. Sawaya conceived and designed the experiments, authored or reviewed drafts of the article, and approved the final draft.

### Data Availability

The data is available at Figshare: Almeida-Silva, Diego; Servino, Leonardo Matheus; Pontes-Nogueira, Matheus; Sawaya, Ricardo (2023). Supplementary Information from

"Marine introgressions and Andean uplift have driven diversification in neotropical Monkey tree frogs (Anura, Phyllomedusinae)". figshare. Dataset. https://doi.org/10.6084/m9.figshare.24282592.v1.

The third-party data used in this study is available at GBIF: https://www.gbif.org/species/4817115.

## Supplemental Information

Supplemental information for this article can be found online at http://dx.doi.org/10.7717/peerj.17232#supplemental-information.

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
