# Peer review of "Marine introgressions and Andean uplift have driven diversification in neotropical Monkey tree frogs (Anura, Phyllomedusinae)"

_PeerJ, doi:10.7717/peerj.17232_

## Round 0.1 · original submission · Major Revisions

Dear authors,

There are a number of requests for clarification from both the reviewers and myself on a range of points ​— addressing these, I feel, will greatly improve the manuscript. Reviewer 3 also suggests a possible re-analysis, and please give this much consideration. I have selected a major revision given that a re-analysis may be necessary, and possible further reconsideration of the discussion, but for the most part, the reviewers have asked for further clarity and justification (which will, as stated above, improve the manuscript for readers).

·

Basic reporting

PeerJ manuscript 91653
In their manuscript, Almeida-Silva et al. reconstruct the phylogeographical history of the subfamily Phyllomedusinae, with particular attention to competing hypotheses concerning their origin, potential vicariance and dispersal.

I need to start this review by declaring that I have no specialist knowledge of monkey tree frogs or their phylogeny. Likewise, I know little of the phylogeography of this area.

The manuscript (ms) is well written and presented, albeit with a large number of annoying grammatical errors that currently spoil the overall quality. There are so many errors that I have chosen to upload a marked up version of the pdf, but I can’t promise that I caught all of the errors, and so urge the authors to seek some help with their grammar.

There is a lot to like in this ms, including the overall approach, analysis and presentation of the results. The minor grammatical errors should be easy to fix given the right help. I am most concerned by some points in the ms where the methods get rather subjective without any justification for what appear to be rather arbitrary decisions. I feel sure that the authors can back up why they chose what they did, but I am also concerned about how robust their results would have been had they made another choice. This requires some substantiation from the authors, both in a rebuttal, but more importantly within the ms to assure readers that the results are robust.

Experimental design

na

Validity of the findings

L126-8 & L133: Here the authors appear to be cherry picking sequences for their phylogeny without much rationalization. Frost is not the best description of species descriptions available. I have always found it rather vague. It would be best here if the authors were to take a systematic approach by providing a list of sequences that were not included and a rationalization on why they were removed (taxonomy, distribution, etc.). This would give more credence to their resulting phylogeny and would provide other workers more reasons to use it.

L189-190: My experience of GBIF is that the records range from good to terrible, and that the taxonomy is often very poor and as outdated as the age of the records. I would add that I have no idea how well it performs for monkey tree frogs. However, I would feel very nervous about accepting all of these records without any filtering (as presented in the ms) and so I ask the authors to provide some explanation about why they had confidence in all records, or how they filtered out bad ones.

L202-3: Can you provide some justification about the choice of 12 (and not 13 or 19)?

L232: see above comment about Frost’s distribution records. How do we know that the results are robust despite this revision?

My major concerns then are that the way in which the authors have written their text implies that at some critical points they have made a bunch of arbitrary decisions that may or may not have influenced the outcome of their study. I am sure that they have very good reasons for these choices, but they need to document these sufficiently for the reader.

Reviewer 2 ·

Basic reporting

The study uses molecular data to infer the history of the Phyllomedusinae clade. In general, the study is very interesting and provide a good understanding of the colonization and diversification into the clade

Experimental design

The authors used molecular data available in GenBank and several statistical techniques to infer the age of clades and infer the historical processes that shaped the evolution of the clade. I think all the analysis is well done and give a reliable hypothesis for the evolution of the clade.

Validity of the findings

The data can be used for several other studies to validate other hypothesis in Hylidae clade. Also, it can be compared to ohter studies and give a big picture of the historical processes that shaped the neotropical fauna along the time.

Additional comments

I think the manuscript worth to be published and I have only a few comments. Specially in the introduction, I think the style of authors could better connect the paragraphs. In the way it is, seems the subjects are just placed together, and does not fit well in a continuity.

Reviewer 3 ·

Basic reporting

There are no major comments on the structure and writing of the manuscript. Overall, the text is cohesive and coherent.

Experimental design

Material and Methods, lines 153–156: This dichotomy between nuclear and mitochondrial fragments is mistaken since what most affects the alignment is the presence of secondary structure. For coding fragments (i.e., all nuclear, cyt-b, and ND1), the implementation of the automatic strategy is sufficient for obtaining robust results. For rDNA fragments (i.e., 12S and 16S) and tDNA, the best strategy implemented by MAFFT is Q-INS-i (https://doi.org/10.1093/bib/bbn013), which considers secondary structure in establishing primary homologies. Therefore, redo the alignments before proceeding with obtaining the phylogenetic hypothesis.

Material and Methods, lines 158–159: Provide details on which partitions were imputed a priori for the program.

Material and Methods, lines 164–165: In which taxon did you root the tree? There is no representative out of Hylidae in your sampling. Include an appropriate root and redo the analyses.

Material and Methods, lines 164–165: Why did you choose concatenation instead of estimating a coalescent species tree? Please provide theoretical justification for your choice, since concatenation tends to bias the analysis towards genes with a higher proportion of variation (i.e., mitochondrial genes; https://doi.org/10.1111/nyas.12747), resulting in overestimated estimates of divergence times.

Material and Methods, lines 171–177: Feng et al. (2017; https://doi.org/10.1073/pnas.1704632114), based on a dating analysis with multiple fossil calibrations, provide estimates of divergence times between Phyllomedusa and Agalychnis, Pelodryadinae and Phyllomedusinae (quite concordant with the date obtained by Hime et al., 2021), between Hylinae and the other two subfamilies of Hylidae, and among various genera of Hylidae that you sampled as outgroups. Therefore, after realigning your sequences, date your tree with these estimates instead of using rates obtained from the literature.

Material and Methods, lines 188–193: Why did you obtain this data? And what did you consider as uncertain records? Detail these questions in this section of the methodology.

Material and Methods, lines 228–232: I haven't checked all distributions, but at least in Pithecopus, there are some errors in the geographical distribution of species. For example, P. ayeaye does not occur in the Northern Atlantic Forest and certainly occurs in the Cerrado (https://doi.org/10.1002/ece3.3261). Pithecopus araguaius occurs in the Southern Amazonia and Western Amazonia, and P. azureus only occurs in the Chaco/Pampa (https://doi.org/10.1186/1471-2156-14-70; https://doi.org/10.1016/j.ympev.2023.107959). I understand that part of these errors may result from Linnean and Wallacean shortfalls, but if they are widespread in this work, the biogeographic reconstruction becomes completely implausible. I suggest that the authors redo the estimates of the geographical distributions of species based on scientific literature research with validated records instead of relying solely on GBIF, as some of the data there result from incorrect taxonomic identifications.

Material and Methods, lines 245–247: Explain why the fact that the BAYAREALIKE model does not consider vicariance is a problem in testing your hypothesis.

Validity of the findings

Abstract, lines 28–30: Recent evidence suggests that the closure of the Isthmus of Panama occurred between ~ 10 (https://doi.org/10.1126/science.aaa2815) and 2.8 Ma (https://doi.org/10.1126/sciadv.1600883) . Therefore, it is implausible that the hypothetical ancestor of Phyllomedusinae was present in Central America 38.6 Ma ago. Please review this information.

Introduction, lines 71–72: Please move the "e.g." to the beginning of the citations.

Introduction, lines 76–77: It is important to mention not only that the subfamily has a broad distribution in the Neotropical region but also that it constitutes an autochthonous endemism concerning it (https://doi.org/10.1111/j.1096-0031.2009.00287.x).

Introduction, lines 77–79: Please provide more details on this geographical distribution, including the latitudinal boundaries and the biomes in which the subfamily is distributed.

Introduction, lines 92–94: Remove that sentence, as it has little relevance to the context of the study.

Introduction, 101–105: Explain in more detail why this hypothesis of south-to-north diversification, based on the arrival of other organisms via Antarctica, could apply to Phyllomedusinae. Why didn't the ancestor of this group emerge in the central, northern, or any other region of South America? The estimated time for the separation between South America and Australia is ~30 Ma (https://doi.org/10.1080/10635150490423430), while recent dating suggests that the mean tMRCA for Phyllomedusinae is more recent than that (https://doi.org/10.1016/j.ympev.2023.107907). In summary, you need to provide a more robust background knowledge and alternative hypotheses in the introduction.

Results, line 282: This result doesn't make sense, as the Isthmus of Panama (unit A) was not formed at that time.

Results, lines 290–293: Here again, the results do not make sense. If there was an archipelago where the Isthmus of Panama is today, as depicted in Figure 4, then jump-dispersal is the only plausible process that would explain the colonization of Agalychnis on those islands.

Discussion, line 371: It is not true. See https://doi.org/10.1016/j.ympev.2023.107907

Discussion, lines 398–399: This older dating may be biased by the tree estimation method (i.e., concatenation) and the calibration method. I strongly propose that the authors redo the analyses, as suggested, and reinterpret the diversification process of the subfamily in light of the new results.

Discussion, lines 427–428: Provide Newick notations, more suitable for textual mention of clades.

Discussion, lines 447–451: If, after redoing the analyses, the results are similar to those presented here, consider citing https://doi.org/10.1016/j.ympev.2023.107959, as they perform a biogeographic reconstruction consistent with the hypothesis of the emergence of at least one group of Pithecopus in the western Amazon.

Discussion, line 464: Replace Magalhães, 2012 by https://doi.org/10.1111/jbi.14578

Figures 3 and 4: Please provide a color legend. Figures and tables should be self-explanatory.

Supplementary Data S1: Remove the filters and color markings in some cells of the spreadsheet. Configure the headers and add captions and a metadata tab. Finally, remove the 'Summary' tab.

Additional comments

My main concern with the article is that the estimate of events associated with the Phyllomedusinae of Central America does not align with the geological history of this area. Additionally, there are several methodological issues in estimating the phylogenetic tree and geographical distributions of species that may have influenced the results.

---

## Round 0.2 · Minor Revisions

Dear authors,

Thank you for your extensive work on the manuscript. The re-analysis and follow-up work have greatly improved the study.

There are a few exceptionally minor edits that I have suggested in the annotated PDF. They should be very quick to deal with. I have given the decision "minor revisions" just so that you have an opportunity to make those edits, but please know that I consider this manuscript "Accepted" (and shall assign it that decision on your next resubmission).

Kind regards,
Prof. Alastair Potts

---

## Round 0.3 · accepted · Accept

Thank you very much for the recent minor corrections.